# Parameters for Calcium Metabolism in Women with Polycystic Ovary Syndrome Who Undergo Stimulation with Letrozole: A Prospective Cohort Study

**DOI:** 10.3390/jcm11092597

**Published:** 2022-05-05

**Authors:** Iris Holzer, John Preston Parry, Klara Beitl, Boban Pozderovic, Rodrig Marculescu, Johannes Ott

**Affiliations:** 1Clinical Division of Gynecologic Endocrinology and Reproductive Medicine, Medical University of Vienna, 1090 Vienna, Austria; iris.holzer@meduniwien.ac.at (I.H.); klara.beitl@meduniwien.ac.at (K.B.); n1613782@students.meduniwien.ac.at (B.P.); 2Parryscope and Positive Steps Fertility, Madison, WI 39110, USA; drprestonparry@gmail.com; 3Department of Obstetrics and Gynecology, University of Mississippi Medical Center, Jackson, MS 39216, USA; 4Department of Laboratory Medicine, Medical University of Vienna, Spitalgasse 23, 1090 Vienna, Austria; rodrig.marculescu@meduniwien.ac.at

**Keywords:** polycystic ovarian syndrome, ovarian stimulation, letrozole, calcium metabolism, vitamin D

## Abstract

For women with polycystic ovarian syndrome (PCOS) and infertility, stimulation with the aromatase-inhibitor letrozole has been recommended as a first-line for ovulation induction. Calcium-associated signaling has also been a component for other ovulation induction and superovulation medications. This study’s aim was to evaluate parameters of calcium metabolism in PCOS women. In a prospective cohort study, 61 anovulatory, infertile PCOS patients who underwent letrozole stimulation were included. Outcome measures were: follicular maturation after letrozole stimulation; parathyroid hormone (PTH); 25-hydroxyvitamin D3 (25OHD3); serum levels of calcium, phosphorus, magnesium, albumin, and total protein. Successful recruitment of a dominant follicle was achieved in 35 patients (57.4%). Women with and without successful follicular development did not differ in serum levels of PTH (38.4 ± 19.7 vs. 39.6 ± 16.2 pg/mL), 25OHD3 (62.5 ± 32.1 vs. 65.4 ± 30.9 nmol/L), calcium (2.36 ± 0.08 vs. 2.37 ± 0.12 mmol/L), or protein (70.2 ± 13.3 vs. 74.0 ± 3.7 g/L), respectively (*p* > 0.05). However, women who were not responsive to letrozole for ovulation induction demonstrated higher anti-Müllerian hormone (AMH) levels (9.7 ± 4.7 vs. 5.0 ± 3.2 ng/mL, *p* = 0.005). In conclusion, the success of letrozole stimulation in women with PCOS is independent from calcium metabolism parameters. However, AMH levels seem predictive of medication resistance.

## 1. Introduction

Polycystic ovary syndrome (PCOS) is one of the most common endocrinopathies of women in their reproductive years and a primary cause of infertility, mainly due to ovulatory dysfunction [1,2]. Its diagnosis is based on hyperandrogenism, oligo-ovulation with associated oligomenorrhea and polycystic ovaries on ultrasound [3,4]. The selective estrogen-receptor modulator clomiphene citrate (CC) used to be the first-line treatment for oligoovulatory and anovulatory women with PCOS. However, treatment with the aromatase-inhibitor letrozole results in higher live-birth and ovulation rates [5,6,7]. As a result, letrozole is now recommended by the International PCOS Network as the first-line ovulation induction agent for women with PCOS and infertility [8].

Evidence suggests 25-hydroxyvitamin D3 (25OHD3) is associated with success after clomiphene stimulation for PCOS [9]. Data are scarce for how letrozole and favorable outcomes overlap with 25OHD3 levels. Although one might argue that the influence of calcium metabolism should be the same despite the use of different stimulation agents, this may not be the case. Moreover, there is an ongoing debate as to whether vitamin D affects anti-Müllerian hormone (AMH) levels, which are elevated with PCOS [10,11].

Concerning women with PCOS, there is potentially an interaction between bone-derived markers and downstream metabolic and hormonal imbalances [12]. A high prevalence of vitamin D deficiency and even an inverse association with insulin sensitivity markers have been shown for PCOS patients [13,14]. Vitamin D supplementation has been shown to increase insulin sensitivity [10] and to decrease androgen levels in women with PCOS and vitamin D deficiency, but does not seem to have these effects in women without PCOS that have vitamin D deficiency [15]. Several theories propose linking vitamin D deficiency and PCOS including the fact that Vitamin D improves insulin action by upregulating the expression of the insulin receptor and enhancing insulin responsiveness for glucose transport. Furthermore, 1.25 (OH)2D3 activates the transcription of the specific nuclear receptor VDR of the human insulin gene and vitamin D regulates intracellular and extracellular calcium, which is important for insulin-mediated actions in insulin responsive tissues. Another factor could be the anti-inflammatory actions of vitamin D [16]. Moreover, it has been demonstrated that women with PCOS have significantly lower serum 25OHD3 levels in comparison to fertile controls [14]. In addition, the serum levels of parathyroid hormone (PTH) and phosphorus are elevated in women with PCOS. In the same study cohort of patients who underwent CC stimulation, a 25OHD3 deficiency was an independent predictive parameter of follicular development and pregnancy [9]. Furthermore, in a large review about female fertility and vitamin D, a positive relationship between a higher vitamin D status and in-vitro-fertilization (IVF) outcome has been seen [16,17]. Polymorphisms in vitamin binding protein (VDBP) gene can be a significant risk factor in many diseases including cancers, and obesity and VDBP might play an important role in the regulation of availability of active fractions of 25(OH)D in PCOS women as well. It was demonstrated that despite lower total 25(OH)D in obese women with PCOS, all women with PCOS (lean and obese) had comparable free and bioavailable 25(OH)D levels, which might be a result of concomitantly lowered serum VDBP levels in obese PCOS women [18,19].

Given the association of vitamin D and calcium metabolism not only with PCOS, but also with clomiphene responsiveness and IVF outcomes, the aim of our study was to evaluate whether parameters of calcium metabolism are also associated with stimulation outcomes after letrozole use for PCOS.

## 2. Materials and Methods

### 2.1. Patient Population

This prospective cohort study was conducted at the Clinical Division of Gynaecologic Endocrinology and Reproductive Medicine of the Medical University of Vienna, Austria. From June 2019 to April 2021, 61 women with anovulatory PCOS and primary and secondary infertility were included, who underwent their first cycle of letrozole stimulation. PCOS was diagnosed in accordance with the criteria of the revised European Society of Human Reproduction and Embryology (ESHRE) and the American Society for Reproductive medicine (ASRM) of 2004 [20]. The following exclusion criteria were applied: previous ovarian stimulation or laparoscopic ovarian drilling; hyperprolactinemia, manifest hyperparathyroidism, previous thyroid or parathyroid surgery (since normal parathyroid hormone (PTH) levels do not confirm a normal postoperative parathyroid function in these patients) [21]; and any additional factors for infertility including male factor, tubal occlusion or uterine fibroids impinging on the uterine cavity.

The study was approved by the Ethics Committee of the Medical University of Vienna (IRB number 1546/2019) and was conducted in accordance with the Declaration of Helsinki and the guidelines of Good Clinical Practice. Written informed consent was obtained for all cases. All records were anonymized and de-identified prior to the analyses.

### 2.2. Outcome Parameters

Follicular development was measured only for the first cycle of letrozole treatment, where the primary outcome was obtaining a mature follicle. Additional outcome parameters were age, body mass index (BMI), obesity defined as BMI > 25 kg/m^2^, characteristics of infertility (duration and primary vs. secondary), pretreatment with myoinositol and metformin, presence of polycystic ovarian morphology on ultrasound (antral follicle number > 12, measured on a Voluson S8TM, (GE Healthcare Austria GmbH & Co OG, Zipf, Austria) which is in accordance with the recent PCOS recommendations [22]), parameters of calcium metabolism (namely PTH and 25-hydroxyvitamin D3 (25OHD3), serum calcium, serum phosphorus, the serum calcium-phosphorus product, serum magnesium, serum albumin, and total serum protein), insulin resistance (defined by a HOMA-IR >2.5, when HOMA-IR = fasting insulin × fasting glucose/22.5 × 18), total testosterone, luteinizing hormone (LH), follicle stimulating hormone (FSH), the LH:FSH ratio, and AMH.

All serum parameters were determined at the Department of Laboratory Medicine, Medical University of Vienna, according to ISO 15,189 quality standards. Cobas electrochemiluminescence immunoassays (ECLIA) were performed on Cobas e 602 analyzers (Roche, Mannheim, Germany) for the determination of serum estradiol, follicle-stimulating hormone (FSH), luteinizing hormone (LH), anti-Mullerian hormone (AMH), testosterone and sex hormone binding globuline (SHBG) as reported previously [23,24]. Serum calcium, phosphorus and magnesium levels were determined photometrically on a Cobas 8000 analyzer (Roche Diagnostics, Rotkreuz, Switzerland). An albumin adjustment of calcium levels was not performed, as all patients had normal albumin levels [25]. PTH and 25-hydroxyvitamin D3 (25OHD3) were measured by electrochemiluminescence immunoassays (Roche Diagnostics, Switzerland). Baseline blood sampling was performed on the second to the fifth day of the menstruation cycle after induction with oral administration of 10 mg dydrogesterone twice for 10 days.

### 2.3. Stimulation Protocol

Letrozole stimulation was performed with a dose of 2.5 mg for 5 days from the fifth to the ninth day of the menstrual cycle. Afterwards, all women underwent a vaginal ultrasound for the purpose of follicle monitoring. Starting with the 10th day of the cycle, all patients underwent examinations with vaginal sonography every second day. As soon as a large follicle with a minimum diameter of ≥18 mm was present on ultrasound, ovulation was induced using 5000 units of human chorionic gonadotropin (HCG). Three days after inducing follicular rupture, oral dydrogesterone 10 mg twice was prescribed for 12 days until a pregnancy test was performed.

### 2.4. Sample Size Calculation

According to a study on calcium metabolism parameters and success of clomiphene stimulation in PCOS women which had been conducted at our department previously [9], a difference in the proportions of women with a 25OHD3 < 25 nmol/L between patients with and without ovulation of about 40% was assumed. The sample size was calculated with a power of 80% and an alpha of 0.05. Thus, a minimal number of 23 patients would be needed per group. Assuming a drop-out rate of 15%, three additional patients were considered necessary for each group. However, given that about 43% of PCOS women achieve ovulation with the use of letrozole [26], when estimating sample size, the smaller group (i.e., the group of women with ovulation) would require 26 patients (43%), whereas the group less likely to ovulate would need to include 35 patients (57%). This resulted in a total sample size of 61 patients.

### 2.5. Statistical Analysis

Statistical analysis was conducted using SPSS 26.0TM (IBM Corp., Armonk, NY, USA). All numeric data were distributed normally and, thus, were reported as means and standard deviation. Categorical data were reported as numbers (*n*) and frequencies (%). Correlation analyses between parameters of calcium metabolism were performed using Pearson’s correlations. *p*-values and correlation coefficients r are provided. Predictive parameters for the main outcome parameter “follicle maturation after letrozole stimulation” were tested using a univariate followed by a multivariate binary regression model. Only parameters reaching significance in the univariate mode were entered into the multivariate approach. For these models, crude and adjusted odds ratios (OR) with their according 95% confidence intervals (95%CI) and *p*-values are provided. Differences were considered statistically significant if *p* < 0.05.

## 3. Results

Basic characteristics of the 61 PCOS patients are shown in Table 1. In the whole study population, significant negative correlations were found for serum calcium and PTH (r = −0.336; *p* = 0.024), 25OHD and PTH (r = −0.286; *p* = 0.048) and also for BMI and 25OHD3 (r = −0.370; *p* = 0.004). Serum calcium, PTH, and 25OHD3 were not significantly correlated with either serum testosterone or AMH (*p* > 0.05, data not shown).

Follicular maturation was achieved in 35 patients (57.4%), whereas this was not the case in 26 women (42.6%). In the univariate binary regression models, only a longer duration of infertility, absence of polycystic ovarian morphology, and lower AMH serum levels were associated with obtaining a dominant follicle after letrozole stimulation. When the three factors were entered into a multivariate model, only AMH remained significant (follicular maturation: 5.0 ± 3.2 vs. a lack of a dominant follicle: 9.7 ± 4.7 ng/mL, adjusted OR 0.76, 95%CI: 0.63–0.92; *p* = 0.005). Notably, none of the parameters of bone metabolism were statistically significant. Details are provided in Table 2.

## 4. Discussion

This prospective cohort study did not detect an association among parameters of calcium metabolism and the outcome of ovarian stimulation with letrozole for anovulatory PCOS women. However, the typical negative correlations of serum calcium and PTH, 25OHD3 and PTH [27], as well as BMI, and 25OHD [28,29] were found.

Despite the well-recognized association between a higher BMI and lower 25OHD3 levels, which was also found in our data, one has to address that the mean BMI of 26.2 kg/m^2^ was lower than usual for infertile anovulatory European women with PCOS. This is of note, since it has been suggested that obese women would benefit more from letrozole stimulation than with clomiphene citrate stimulation [6,29]. To the best of our knowledge, the influence of calcium metabolism on stimulation success in PCOS women has only been demonstrated for clomiphene citrate [9]. Although evidence suggests that vitamin D might have beneficial effects on hormonal parameters of PCOS and appears to be associated with IVF outcome, no cause–effect relationship has been established so far [16,17]. However, there is a study where patients receiving supplementation with calcium and vitamin D in addition to metformin seemed to have more rapid follicular progression than women who were treated with metformin only [28].

Thus, our data about PCOS women who underwent ovarian stimulation with 2.5 mg letrozole stands in contrast to the above-mentioned prospective observational study that demonstrated an association of low 25OHD3 levels and 25OHD3 deficiency <25 nmol/L and lower rates of follicle development and pregnancy rates after stimulation with 50 mg CC [9]. The causes for the discrepancy between a stimulation with CC and letrozole can only be hypothesized. First, one could argue that the sample size was insufficient. Although a higher-than-expected rate of follicular maturation was achieved, the smaller group consisted of exactly 26 patients as indicated through power calculations. Moreover, with no dropouts, the necessary minimum number of 23 patients per group was even exceeded. Second, it should be noted that in the cohort, the mean serum 25OHD3 levels were comparably high (follicle maturation: 62.5 ± 32.125 nmol/L vs. no follicle maturation: 65.4 ± 30.9mol/L), which is also reflected by the fact that only six women (9.8%) revealed 25OHD3 deficiency <25 nmol/L (Table 2). Thus, in contrast to other PCOS patients, vitamin D deficiency was not common among our patients [14]. This circumstance could be linked to the only moderate mean BMI of 26.2 kg/m^2^. One potential associated explanation would be the healthy user effect, where populations with vitamin D deficiency were more likely to have concurrent comorbidities actually driving their anovulation and the association was simply confounding. To explore molecular mechanisms in detail, the above-mentioned association between 25OHD3 deficiency and obesity has been suggested to be due to lipophilic sequestration of the vitamin in adipose tissue as well as lower sunlight exposures in obese subjects [30,31]. However, a randomized controlled trial about 25OHD3 serum concentrations below 75 mmol/L in patients with PCOS could not show any significant effect of a supplementation with vitamin D on metabolic or endocrine parameters. In addition, despite the vitamin D supplementation, data were unable to show an improvement in menstrual frequency [32]. The lack of interventional efficacy is likely further evidence of healthy user effects on serum levels, rather than direct causation. Accordingly, we suspect our main findings may be due to the absence or an only minor influence of calcium metabolism and letrozole stimulation success on ovulation induction, but further research is needed for validation.

A meta-analysis provided evidence that physical activity may improve pregnancy rates in women with reproductive health problems [33]. Furthermore, it was demonstrated that cumulative ovulation rates after standardized ovulation induction with CC were superior after weight loss and lifestyle modification in comparison to oral contraceptive pretreatment [34]. Before starting ovulation induction, the effects of lifestyle modification, physical activity and weight loss in case of obesityshould be explained to all patients to accomplish better pregnancy rates.

Apart from 25OHD3, follicular maturation was not associated with PTH, calcium, magnesium, phosphorus, or serum protein levels. These observations are consistent with previous studies. It has been demonstrated that PTH and serum calcium were not predictive parameters for CC stimulation outcome in PCOS women [9]. In a study on women undergoing stimulation with clomiphene and human-menopausal gonadotrophin, it could be shown that the levels of calcium, copper, and zinc in 33 laparoscopically collected follicular fluid did not significantly differ according to parameters of follicle quality, namely size, presence of an oocyte, and presence of a fertilizable oocyte. The authors concluded that calcium levels are not predictive of follicular development [35].

Notably, our study confirmed that AMH levels, duration of infertility, and a polycystic ovarian morphology (PCOM) are factors associated with a follicle maturation in women diagnosed with PCOS undergoing stimulation with letrozole. The predictive value of serum AMH in women with PCOS undergoing CC stimulation is already known indicating that PCOS women with high circulating AMH are often resistant to CC and may require a higher starter dose [36,37]. Similar findings have already been reported for letrozole stimulation in PCOS women [38,39,40,41]. The association between AMH levels and polycystic ovarian morphology on ultrasound is well-recognized [42] and consistent with our multivariate model, where only AMH remained a statistically significant predictor for follicle maturation (Table 2).

Our study is limited by the lack of a pregnancy or live birth rate as outcome parameters. In addition, we did not provide data about the patients’ ethnicity and the physical activity rate that might play a role in letrozole responsiveness as well. Moreover, evaluation of vitamin D binding protein levels and free vitamin D levels might have been interesting. The lack of these data must also be considered a study limitation.

However, we consider our sample size sufficient to explore relevant associations between parameters of calcium metabolism and follicle maturation after letrozole stimulation. If an association among these serum levels and ovulatory patterns cannot be readily established, the sample size required to demonstrate an effect on pregnancy and live birth outcomes is likely enormous. Moreover, the fact that established parameters such as AMH and the presence of polycystic ovarian morphology were significantly predictive, point towards the reliability of our data set.

## 5. Conclusions

The efficacy of ovulation induction through letrozole 2.5 mg for five days in infertile, anovulatory PCOS women is associated with lower serum AMH levels, whereas parameters of calcium metabolism, such as serum calcium, 25OHD3, and PTH do not seem to be predictive. In addition, it would be desirable to evaluate whether calcium and or vitamin D supplementation, two measures that are performed frequently in clinical routine, are capable of improving stimulation outcomes in PCOS women.

## Figures and Tables

**Table 1 jcm-11-02597-t001:** Basic patient characteristics and basic hormonal profile of the 61 anovulatory, infertile PCOS patients.

Age (years) ^1^	28.0 ± 4.2
BMI (kg/m2) ^1^	26.2 ± 5.0
Obesity ^2^	13 (21.3)
Duration of infertility (months) ^1^	31.0 ± 20.4
Secondary infertility ^2^	16 (26.2)
Number of previous pregnancies ^2^	
0	45 (73.8)
1	12 (19.7)
≥2	4 (6.5)
Insulin resistance ^2^	14 (23.0)
Metformin treatment ^2^	11 (18.0)
Myoinositol treatment ^2^	31 (50.8)
Polycystic ovarian morphology ^2^	47 (77.0)
LH (IU/mL) ^1^	11.1 ± 5.8
FSH (IU/mL) ^1^	6.1 ± 1.4
LH to FSH ratio ^1^	1.9 ± 1.1
Testosterone (ng/mL) ^1^	0.41 ± 0.21
AMH (ng/mL) ^1^	7.0 ± 4.5

Data are provided as ^1^ mean ± standard deviation or ^2^ number (frequency), PCOS = polycystic ovary syndrome, BMI = body mass index, LH = luteinizing hormone, FSH = follicle stimulating hormone, AMH = anti-Müllerian hormone.

**Table 2 jcm-11-02597-t002:** Univariate followed by multivariate binary regression model for the prediction of follicle maturation after letrozole stimulation.

Parameter	Letrozole Responsive Patients (*n* = 35)	Letrozole Resistant Patients (*n* = 26)	Crude OR (95%CI)	*p*	Adjusted OR (95% CI)	*P*
Age (years) ^1^	28.1 ± 4.5	27.8 ± 3.8	1.02 (0.90; 1.15)	0.776	-	-
BMI (kg/m^2^) ^1^	26.5 ± 4.4	25.8 ± 5.7	1.03 (0.93; 1.14)	0.610	-	-
Obesity ^2^	8 (22.9)	5 (19.2)	1.24 (0.36; 4.37)	0.733	-	-
Duration of infertility (months) ^1^	35.8 ± 21.2	24.4 ± 17.5	1.03 (1.00; 1.06)	0.037	1.02 (0.98; 1.05)	0.386
Secondary infertility ^2^	8 (22.9)	8 (30.8)	0.67 (0.21; 2.1)	0.488	-	-
Insulin resistance ^2^	7 (20.0)	7 (26.9)	0.68 (0.21; 2.25)	0.526	-	-
Myoinositol treatment ^2^	14 (40.0)	17 (65.4)	0.353 (0.12; 1.01)	0.053	-	-
Metformin treatment ^2^	6 (17.1)	5 (19.2)	0.869 (0.23; 3.23)	0.834	-	-
Polycystic ovarian morphology ^2^	23 (65.7)	24 (92.3)	0.16 (0.03; 0.79)	0.025	0.30 (0.05; 1.69)	0.172
LH to FSH ratio ^1^	1.8 ± 1.2	2.1 ± 0.8	0.77 (0.47; 1.26)	0.290	-	-
Testosterone (ng/mL) ^1^	0.37 ± 0.18	0.47 ± 0.24	0.08 (0.01; 1.19)	0.066	-	-
AMH (ng/mL) ^1^	5.0 ± 3.2	9.7 ± 4.7	0.73 (0.61; 0.87)	<0.001	0.76 (0.63; 0.92)	0.005
25OHD3 (nmol/L) ^1^	62.5 ± 32.1	65.4 ± 30.9	1.00 (0.98; 1.01)	0.719	-	-
25OHD3 < 25 nmol/L ^2^	3 (8.6)	3 (11.5)	0.72 (0.13; 3.89)	0.701	-	-
Parathyroid hormone (pg/mL) ^1^	38.4 ± 19.7	39.6 ± 16.2	1.00 (0.97; 1.03)	0.825	-	-
Serum calcium (mmol/L) ^1^	2.36 ± 0.08	2.37 ± 0.12	0.23 (0.00; 72.87)	0.614	-	-
Serum protein (g/L) ^1^	70.2 ± 13.3	74.0 ± 3.7	0.89 (0.76; 1.04)	0.154	-	-
Albumin (g/L) ^1^	46.6 ± 2.4	46.9 ± 2.7	0.96 (0.76; 1.20)	0.695	-	-
Magnesium (mmol/L) ^1^	0.80 ± 0.06	0.81 ± 0.06	0.08 (0.00; 806.17)	0.596	-	-
Phosphorus (mmol/L) ^1^	1.04 ± 0.13	1.11 ± 0.16	0.04 (0.01; 2.14)	0.110	-	-
Calcium phosphorus product (mmol^2^/L^2^) ^1^	2.48 ± 0.32	2.62 ± 0.40	0.31 (0.06; 1.62)	0.165	-	-

Data are provided as ^1^ mean ± standard deviation or ^2^ number (frequency), *n* = number, OR = odds ratio, BMI = body mass index, LH = luteinizing hormone, FSH = follicle stimulating hormone, 25OHD3 = 25-hydroxyvitamin D3, AMH = anti-Müllerian hormone.

## Data Availability

Data are available on reasonable request. The data presented in this study are available on request from the corresponding author. The data are not publicly available due to their use in forthcoming publications.

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
