# Peer review of "Parameters for Calcium Metabolism in Women with Polycystic Ovary Syndrome Who Undergo Stimulation with Letrozole: A Prospective Cohort Study"

_jcm, 2022, doi:10.3390/jcm11092597_

Round 1
Reviewer 1 Report
Holzer et al. aimed to evaluate the parameters of calcium metabolism (PTH, vitamin D, calcium...) in 61 anovulatory PCOS patients who underwent letrozole stimulation. The paper is clearly written and concise.
Content:
- The introduction can be expanded to discuss the potential role of vitamin D deficiency in PCOS to emphasize the importance of this research.
- A very similar study was done in 2012 by one of the authors where they investigated the effect of parameters for calcium metabolism in women with polycystic ovary syndrome who undergo clomiphene citrate stimulation. In that study, they showed that 25OHD3 deficiency was an independent predictive parameter of clomiphene stimulation outcome, in terms of follicle development and pregnancy. This makes this paper less interesting and novel. The negative results seen here could be attributed to the low sample size (few participants who had low vitamin D levels). What that study showed is that women with PCOS (and women in general) who had low levels of vitamin D should be supplemented with vitamin D. The conclusion of this study, regardless of the results, will not change the fact that women who have low levels of vitamin D should be encouraged to be supplemented with vitamin D.
- It would be interesting to take a closer look on other vitamin D related metabolites such as vitamin D binding protein and free levels of vitamin D.
- It would also be interesting to see if vitamin D or calcium supplementation enhances the rate of letrozole responsiveness.
Figures and tables:
- Figure 1: I don’t think figure 1 is necessary because the information outlined is already exhaustively established in the literature (significant correlations between the calcium metabolism markers and the relationship between vitamin D and obesity). These results can be mentioned in the results section without the need of a figure. Also, the numbers in the figures are very small and not easily readable, especially the p-value and correlation coefficient (r).
- Table 1 can have more information on baseline characteristics (metformin treatment, previous pregnancies…)
- A table should be added with the biochemical profile of patients before letrozole stimulation if this information is available.
Minor edits:
- Table 2, first row: letrozol was written instead of letrozole
- Line 171: this prospective cohort study did NOT detect an association
- Line 183: no cause-effect relationship has been established
Author Response
Comment 1:
Holzer et al. aimed to evaluate the parameters of calcium metabolism (PTH, vitamin D, calcium...) in 61 anovulatory PCOS patients who underwent letrozole stimulation. The paper is clearly written and concise
The introduction can be expanded to discuss the potential role of vitamin D deficiency in PCOS to emphasize the importance of this research.
A very similar study was done in 2012 by one of the authors where they investigated the effect of parameters for calcium metabolism in women with polycystic ovary syndrome who undergo clomiphene citrate stimulation. In that study, they showed that 25OHD3 deficiency was an independent predictive parameter of clomiphene stimulation outcome, in terms of follicle development and pregnancy. This makes this paper less interesting and novel. The negative results seen here could be attributed to the low sample size (few participants who had low vitamin D levels). What that study showed is that women with PCOS (and women in general) who had low levels of vitamin D should be supplemented with vitamin D. The conclusion of this study, regardless of the results, will not change the fact that women who have low levels of vitamin D should be encouraged to be supplemented with vitamin D.
Reply: We thank the reviewer for this important comment. As suggested, we expanded the introduction to discuss the role of vitamin D deficiency in PCOS. Thus, we included the following section: “Several theories propose linking vitamin D deficiency and PCOS including the fact that Vitamin D improves insulin action by upregulating the expression of the insulin receptor and enhancing insulin responsiveness for glucose transport. Furthermore, 1.25 (OH)2D3 activates the transcription of the specific nuclear receptor VDR of the human insulin gene and vitamin D regulates intracellular and extracellular calcium, which is important for insulin -mediated actions in insulin responsive tissues. Another factor could be the anti-inflammatory actions of vitamin D [16].”
New reference:
- Voulgaris N.; Papanastasiou L.; Piaditis G.; Angelousi A.; Kaltsas G.; Mastorakos G.; Kassi E. Vitamin D and aspects of fe-male fertility. Hormones (Athens) 2017, 16:5-21.
Comment 2:
It would be interesting to take a closer look on other vitamin D related metabolites such as vitamin D binding protein and free levels of vitamin D.
Reply: We thank the reviewer for bringing this important point to our attention. To discuss a potential role of the vitamin D binding protein, we added the following phrases: “Polymorphisms in vitamin binding protein (VDBP) gene can be a significant risk factor in many diseases including cancers and obesity and VDBP might play an important role in the regulation of availability of active fractions of 25(OH)D in PCOS women as well. It was demonstrated that despite lower total 25(OH)D in obese women with PCOS, all women with PCOS (lean and obese) had comparable free and bioavailable 25(OH)D levels, which might be a result of concomitantly lowered serum VDBP levels in obese PCOS women [18,19].“
New references:
- Kuliczkowska-Plaksej J.; Pasquali A.; Milewicz A. Serum Vitamin D Binding Protein level associated with metabolic cardi-ovascular risk factors in women with the Polycystic Ovary Syndrome. Horm Metab Res. 2019, Jan; 51(1):54.61.
- Rozmus D.; Ciesielska A.; Płomiński J.; Grzybowski R.Vitamin D Binding Protein (VDBP) and its gene polymor-phisms-the risk of malignant tumors and other diseases. Int J Mol Sci. 2020 Oct 22;21(21):7822
However, since we report the results of prospective cohort study (rather than a bio-banking project), we cannot provide data about vitamin D binding protein or free vitamin D levels. We address this issue as a study limitation as follows: “Moreover, evaluation of vitamin D binding protein levels and free vitamin D levels might have been interesting. The lack of these data must also be considered a study limitation.”
Comment 3: It would also be interesting to see if vitamin D or calcium supplementation enhances the rate of letrozole responsiveness.
Reply: We agree with Reviewer 1 that data about vitamin D and calcium supplementation and letrozole responsiveness would be of interest. In the revised manuscript, we point out to the fact that this would be of interest for future studies: “In addition, it would be desirable to evaluate whether calcium and or vitamin D supple-mentation, two measures that are done frequently in clinical routine, are capable of im-proving stimulation outcomes in PCOS women. ”
Comment 4: Figures and tables:
Figure 1: I don’t think figure 1 is necessary because the information outlined is already exhaustively established in the literature (significant correlations between the calcium metabolism markers and the relationship between vitamin D and obesity). These results can be mentioned in the results section without the ned of a figure. Also, the numbers in the figures are very small and not easily readable, especially the p-value and correlation coefficient (r).
Reply: We agree. As all information is mentioned in the text, we removed Figure 1 to make the manuscript more easily readable.
Comment 5: Table 1 can have more information on baseline characteristics (metformin treatment, previous pregnancies…
Reply: We added the following baseline data to Table 1: Number of previous pregnancies, concomitant metformin and myo-inositol treatment.
Comment 6: A table should be added with the biochemical profile of patients before letrozole stimulation if this information is available.
Reply: Data about PCOS-typical parameters which include FSH, LH, LH:FSH ratio, testosterone, and AMH were available. These were also added to Table 1.
Comment 7: Minor edits:
Table 2, first row: letrozol was written instead of letrozole
Line 171: this prospective cohort study did NOT detect an association
Line 183: no cause-effect relationship has been established
Reply: All three phrases were corrected in our manuscript.

Reviewer 2 Report
Dear Authors,
The aim of the study under the title: „
Parameters for calcium metabolism in women with Polycystic Ovary Syndrome who undergo stimulation with Letrozole: a prospective cohort study was to evaluate whether parameters of calcium metabolism are associated with stimulation outcomes after letrozole use for PCOS infertile women. The finding is very interesting. I have, however, some minor concerns about the design that needs to be addressed. My comments are listed below.
Abstract
The summary includes all the necessary elements
Introduction
The introduction is well made.
Material and methods
Does patient racial/ethnicity was a matter of the study?
Results
In the footer of tables 1 and 2, the description of the abbreviations used and all units (e.g. frequency) specified in the table are missing
The limitation of the study is the lack of pregnancy or live birth rate, as well as physical activity rate, as outcome parameters.
Discussion and Conclusions
More and more studies show that patients' lifestyle plays a very important role in both PCOS and infertility. Both the research methodology and the discussion lacked information on this subject. Do the factors related to e.g. physical activity could have had an impact on the results obtained?
Author Response
Reviewer #2:
The aim of the study under the title: „Parameters for calcium metabolism in women with Polycystic Ovary Syndrome who undergo stimulation with Letrozole: a prospective cohort study was to evaluate whether parameters of calcium metabolism are associated with stimulation outcomes after letrozole use for PCOS infertile women. The finding is very interesting. I have, however, some minor concerns about the design that needs to be addressed. My comments are listed below.
Reply: We thank the reviewer for his positive overall assessment of our manuscript.
Abstract
Comment 1: The summary includes all the necessary elements
Introduction
Comment 2: The introduction is well made.
Material and methods
Comment 3: Does patient racial/ethnicity was a matter of the study?
Reply: Due du lacking information about patient ethnicity, this variable was not included in our statistical analysis. To make this more clear to the readership we included the following sentence in the limitation section: “ In addition, we did not provide data about the patients ethnicity and the physical activity rate that might play a role in letrozole responsiveness as well.”
Results
Comment 4: In the footer of tables 1 and 2, the description of the abbreviations used and all units (e.g. frequency) specified in the table are missing
Reply: We corrected the following suggestions.
Comment 5: The limitation of the study is the lack of pregnancy or live birth rate, as well as physical activity rate, as outcome parameters.
Reply: We totally agree that beside the lack of pregnancy and live birth rates, the physical activity rate is missing as potential variable. As stated out earlier the following sentence was included in our manuscript: “In addition, we did not provide data about the patients ethnicity and the physical activity rate that might play a role in letrozole responsiveness as well.”
Discussion and Conclusions
Comment 6: More and more studies show that patients' lifestyle plays a very important role in both PCOS and infertility. Both the research methodology and the discussion lacked information on this subject. Do the factors related to e.g. physical activity could have had an impact on the results obtained?
Reply: We share the Reviewers opinion that lifestyle modification and physical activity is of great importance for women with PCOS. Therefore, we encourage all our obese women to lose weight and start physical activity. We decided to start treatment with letrozole only in patients with a BM ≤32 kg/m2. To emphasize the factor lifestyle we added the following phrase in the discussion section: “A meta-analysis provided evidence that physical activity may improve pregnancy rates in women with reproductive health problems [33]. Furthermore, it was demonstrated that cumulative ovulation rates after standardized ovulation induction with CC were superior after weight loss and lifestyle modification in comparison to oral contraceptive pretreatment [34]. Before starting ovulation induction the effects of lifestyle modification, physical activity and gain loss should be explained to all patients to accomplish better pregnancy rates.“
New references:
- Meno, G.P.; Mielke G.I.; brown W., J. The effect of physical activity on reproductive health otcomes in young women: a sys-tematic review and meta-analysis. Hum Reprod Update 2019, 11; 25(5): 541-563
- Legro, R.S.; Dodson W.C.; Kris-Etherton, P.M. Randomized controlled trial of preconception interventions in infertile women with PCOS. J Clin Endocrinol Metab 2015, 100 (11):4048-58.

Round 2
Reviewer 1 Report
The authors appropriately addressed my comments.